# Conjunctival Extramedullary Plasmacytoma in Dogs and a Cat: Clinical Characteristics and Histopathological Findings

**DOI:** 10.3390/ani12213015

**Published:** 2022-11-03

**Authors:** Aleksandra Rawicka, Manuela Crasta, Adolfo Guandalini, Claudio Peruccio, Domenico Multari, Marco Rondena, Nunzio D’Anna, Chiara Giudice

**Affiliations:** 1Clinica Veterinaria Roma Sud, Via Pilade Mazza 24, 00173 Rome, Italy; 2Anicura VisionVet Eye Clinic, Via Antonio Marzocchi 6, San Giovanni in Persiceto, 40017 Bologna, Italy; 3Department of Ophthalmology, Centro Veterinario Specialistico, Via Sandro Giovannini 53, 00137 Rome, Italy; 4Veterinary Ophthalmology Referrals Centro Veterinario Torinese, Lungo Dora Colletta 147, 10153 Torino, Italy; 5Centro Veterinario Oculistico Fontane, Via Donatori del Sangue 1, 31020 Villorba, Italy; 6Clinica Veterinaria Privata San Marco Srl, Via dell’Industria, 3, 35030 Veggiano, Italy; 7Department of Veterinary Medicine and Animal Sciences, Università degli Studi di Milano, 26009 Lodi, Italy

**Keywords:** extramedullary plasmacytoma, conjunctiva, eye, dog, cat

## Abstract

**Simple Summary:**

Extramedullary plasmacytoma has been occasionally reported to affect the ocular and periocular (orbit, conjunctiva) region. To better describe the clinical and histopathological features of extramedullary plasmacytoma, we reviewed the records of seven cases (six dogs, one cat). In these cases, extramedullary plasmacytoma was found mostly in the conjunctiva of the third eyelid in the form of salmon-colored nodules with clear borders. All tumors were surgically removed with no additional therapy. No signs of recurrence or systemic involvement (multiple myeloma) were observed in the follow up period. We recommend that extramedullary plasmacytoma should be considered in cases of canine and feline conjunctival masses, followed by investigations to rule out multiple myeloma. Our results suggest that conjunctival extramedullary plasmacytoma is unlikely to recur or spread, and that local surgical excision alone or combined with cryotherapy should be curative.

**Abstract:**

Background: Extramedullary plasmacytoma (EMP) is a plasma cell tumor that is very rarely reported in ocular and adnexal tissue and is usually solitary and benign. Methods: This study evaluated seven cases from different ophthalmology referrals submitted for histopathological analysis between 2013 and 2022. Results: EMPs were diagnosed in a 9-year-old domestic short-haired cat and in six dogs (median age, 10 years): three English Cocker Spaniels, a Golden Retriever, a Maltese, and a Pinscher. EMPs affected the third eyelid conjunctiva in five cases (5/7), followed by the superior bulbar (1/7) and palpebral conjunctivae (1/7), respectively, and were presented mostly as well-defined, salmon-colored nodules. Histopathologically, the masses were composed of sheets and packets of round cells morphologically consistent with mature plasma cells with mild anisocytosis and anisokaryosis; mitosis and binucleated giant cells were rarely observed. Follow up for 6/7 cases ranged from 6 months to 2 years. No signs of recurrence or systemic involvement (multiple myeloma) were observed. Conclusions: EMP should be included in the differential diagnoses of canine and feline conjunctival masses. Staging recommendations should include multiple myeloma diagnostic evaluation. Our results suggest that primary conjunctival EMP does not appear to be locally aggressive and has low systemic involvement behavior. The local surgical excision alone or combined with cryotherapy should be curative.

## 1. Introduction

Extramedullary plasmacytoma (EMP) is a plasma cell tumor, usually solitary and benign in nature, and mostly non-recurrent following excision [1,2,3]. In a large study of 75 canine EMPs, the most common location was the skin (86%), followed by the mucous membranes of the oral cavity (9%), then the rectum and colon (4%). Other sites, such as the stomach, spleen, genitalia, eye, uterus, and liver accounted for the remaining 1% of the cases [2]. EMPs are less frequently reported in cats. A previous study suggests that, in this species, they originate in soft tissues but later present in bones [4].

The diagnosis of EMP is generally histological, and exclusion of a disseminated lesion is advised. Specifically, monoclonal gammopathy, osteolytic lesions, Bence-Jones (i.e., light chain) proteinuria, or bone marrow plasmacytosis of greater than 5% should be ruled out [5].

Although EMP located in the skin, subcutis, or gastrointestinal tract is widely described in the literature, only individual case reports refer to the “eye”: specifically, the third eyelid [6], orbit [7], or intraocular tissues [8].

Therefore, the aim of the present paper is to describe the clinico-pathological features of conjunctival extramedullary plasmacytoma in six dogs and one cat.

## 2. Materials and Methods

Medical and histopathological records of dogs and cats with conjunctival EMP treated surgically between 2013 and 2022 at five referral ophthalmological centers were reviewed.

The information that was recorded included the breed, age, sex, neuter status, affected eye, previous therapy or concurrent ophthalmic condition, size, localization and clinical features of the tumor, previously performed cytology/biopsy, histopathological diagnosis, additional staging diagnostic tests [e.g., complete blood count (CBC), serum chemistry profile, serum protein electrophoresis, urine analysis, skeletal radiographs, abdominal ultrasound, bone marrow aspirate, computer tomography scan (CT)], and follow up.

A physical evaluation and pre-anesthetic bloodwork were performed in all cases, as well as complete ophthalmic examinations by an ophthalmology resident-in-training under the direct supervision of a board-certified veterinary ophthalmologist.

The ophthalmic examination included slit lamp biomicroscopy (Kowa SL 17^®^, Kowa Company Ltd., Tokyo, Japan) and indirect ophthalmoscopy (Omega 500; Heine, Ettenheim, Germany or Vantage Plus; Keeler, Windsor, UK) with either a 30D or a 2.2 Panretinal lens (Volk Optical Inc., Mentor, OH, USA). The Schirmer tear test I (MSD Animal Health^®^, Madison, NY, USA), rebound tonometry (Icare^®^ TonoVet, Helsinki, Finland), and the fluorescein test (Fluoresceina Fluoro-Touch strip, Madhu Instruments, New Delhi, India) were also performed.

Immediately after surgery, the removed masses were fixed in 10% buffered formalin, and therefore sent to the laboratory, routinely processed for histological examination, and embedded in paraffin. Five-micrometer microtomic sections were stained with hematoxylin and eosin (H–E).

Immunohistochemistry was performed using a Leica Bond RX^®^ (Leica Biosystems, Newcastle, UK) fully automated research stainer with the following antibodies: anti-CD3 (mouse monoclonal, clone LN10, Novocastra^TM^, Newcastle, UK), anti-Lambda chain (rabbit polyclonal, Agilent, Santa Clara, CA, USA), anti-CD20, (rabbit polyclonal, Invitrogen, Waltham, MA, USA), and anti-MUM1 Protein (mouse monoclonal, clone MUM1p, Agilent, Santa Clara, CA, USA).

Briefly, adhesive slides with seriate 4 micrometer thick sections were obtained and stained using an appropriate immunohistochemical protocol based on diaminobenzidine chromogen and counterstained with hematoxylin, using a Leica biosystems^®^ kit (Leica Biosystems, Newcastle, UK). Bond Polymer Refine Detection^®^ (Leica Biosystems, Newcastle, UK) Heat induced epitope retrieval (HIER) was performed for anti-CD3, anti-Lambda chain and anti-MUM1 Protein antibodies, using a pH 6.0 buffer (Bond Epitope Retrieval Solution 1, Leica biosystems, Newcastle, UK) at 99 °C for 20 min.

Sections of hyperplastic canine and feline lymph-node were used as positive controls. For negative controls, the primary antibodies were replaced with mouse or rabbit IgG (Santa Cruz; Dallas, TX, USA, sc-2025 and sc-2027, respectively).

The presence or absence of recurrence and the systemic status of the patients were determined during the subsequent recheck examinations. Additional follow-up information was obtained by telephone or email interviews with the pet owners and/or the referring veterinarians.

## 3. Results

Seven cases (six dogs, one cat) of conjunctival EMP were retrieved from the archive of Pathology of the Department of Veterinary Medicine and Animal Sciences (Appendix A).

The mean age of dogs at treatment was 10 years (range 5–15 years). The patients represented four breeds: English Cocker Spaniel (3/6), Golden Retriever (1/6), Maltese (1/6), and Pinscher (1/6). There were two spayed females and 4 males, three of which were entire. Moreover, one 9-year-old, spayed female, domestic short-hair cat met the inclusion criteria.

Ophthalmic examination revealed smooth, painless, well-defined, raised nodules in all cases (salmon-colored in six patients and brownish in case 2) (Appendix A).

No ocular discharge or bleeding was noted in any case. The diameter of the masses ranged from 2 to 15 mm (mean, 8.3 mm; median, 8.0 mm). Masses were characterized by rapid growth and were localized in the third eyelid (5/7), and the superior bulbar (1/7) and upper eyelid (1/7) conjunctiva. In all cases, the masses were unilateral; three were localized in the left eye (OS), and four in the right (OD).

Surgery was preceded by a cytological examination in case 6 (Appendix A) and incisional biopsy of the mass in case 7. Both of these were diagnosed as conjunctival plasmacytoma.

The results of the physical examination, including palpation of the regional lymph nodes, were within normal limits. CBC, serum chemistry and a urine test were performed in five cases, and no relevant abnormalities were detected, except for a mild increase in serum cholesterol (668 mmol/L reference range 100–300 mmol/L) and triglyceride concentration (303 mmol/L, reference range 26–100 mmol/L) in case 3. Total globulin concentration was evaluated in five animals (four dogs and the cat) and was within normal limits in all. In three patients, serum protein electrophoresis was also performed as a screening test, and the results were all normal. No patient had confirmed monoclonal gammopathy. Urine protein electrophoresis was performed as a screening test in three patients with normal results in all three (Appendix A).

The neoplastic lesions were surgically removed in all cases, sparing the surrounding eye structures such as the third eyelid; in case 7, surgical excision was combined with cryotherapy.

Histologically, all canine cases showed similar findings. The neoplasms were composed of round cells organized in dense sheets and nests sustained by delicate fibrovascular stroma. In all cases, neoplastic cells, despite some degree of nuclear pleomorphism, resembled plasma cells, with moderate eosinophilic to amphophilic cytoplasm (10 to 20 µm in diameter) occasionally with a distinct perinuclear clear zone (Golgi apparatus) and a round nucleus with finely-stippled, or more rarely, clockface chromatin. The mitotic count varied from two to nine mitoses in 10 high power fields (HPFs) (2.37 mm^2^). Anisocytosis and anisokaryosis were generally moderate, with scattered macrokaryosis and binucleated cells, and only occasional tri-nucleated cells (case 3). In case 7, neoplastic cells were intermingled with large deposits of amyloid (eosinophilic fibrillar ground substance Congo Red +) (Appendix A). The neoplasms extended to the cut borders in three cases (cases 3, 4, 7).

In the cat, a poorly defined neoplastic mass, 1.5 cm in diameter, expanded the conjunctiva of the third eyelid, infiltrating and replacing the third eyelid glandular tissue, not reaching the cut borders. The neoplasm was composed of dense sheets of round cells morphologically consistent with plasma cells. Anisocytosis and anisokaryosis were moderate, with rare, scattered giant binucleated cells. Mitoses ranged from zero to two per HPF.

Immunohistochemically, neoplastic cells in all canine cases were diffusely positive for MUM 1 and negative for CD3. Scattered cells were CD20+ positive cells. In the cat, neoplastic cells were diffusely positive for lambda chains and negative for CD3, with only a few scattered CD20+ positive cells (Appendix A).

CT or abdominal ultrasound with radiographic survey, to exclude disseminated lesions and bone lysis, were performed in three cases (two dogs, one cat). Changes suggestive of multiple myeloma (MM) were not reported in any case. In case 1, CT revealed bilateral mild stenosis of the external auditory canal, absence of the left testicle, diffuse bronchopathy, and prostatic hypoplasia. In case 7, CT revealed a splenic nodule that was cytologically consistent with lymphoid hyperplasia.

In case 7 and in the feline patient, bone marrow aspiration was performed. In both cases, MM was excluded.

Based on these clinical, histological, and immunophenotypical findings, EMP was diagnosed in all seven patients.

No cases received additional postsurgical therapies for plasmacytoma. In two cases (n. 1, n. 3) that were diagnosed with kerato-conjunctivitis sicca when referred for the conjunctival mass, treatment with cyclosporine was started after surgery.

Follow up data were available for six cases (five dogs, one cat); follow up time varied from 6 months to 2 years. No symptoms of recurrence or systemic signs of MM were observed.

## 4. Discussion

This case series describes clinical and histopathological features of primary conjunctival EMPs in six dogs and one cat.

In humans, conjunctival plasmacytomas are extremely rare [9] and mostly originate from the extension of orbital lesions. Most human orbital plasmacytomas are diagnosed in patients with a history of MM or in asymptomatic cases of MM where the diagnosis is not yet established [10]. For example, Thuro et al. evaluated 30 patients with orbital plasmacytoma: the majority (63%) were already diagnosed with MM, while 37% were diagnosed immediately after the identification of the orbital plasmacytoma [11]. Primary conjunctival EMPs are very unusual in human beings: only six cases have been reported so far in the literature, affecting patients from 17 to 66 years of age [10,12,13,14,15,16].

A single case of conjunctival/third eyelid gland primary EMP has previously been reported in a dog [6]. Additionally, presumed intraocular [8] and orbital [9] primary EMPs have been described in a cat and a dog, respectively. In both of these species, cutaneous and gastrointestinal EMPs are more common and occur mainly in middle aged to older animals with no apparent sex predisposition [17], but with a higher occurrence in some dog breeds: American and English Cocker Spaniels, Airedale, Kerry Blue and Scottish Terriers [2,18].

In our small case series of primary conjunctival EMPs, Cocker Spaniels were overrepresented (3/6), and in the previously published case report of conjunctival/third eyelid gland plasmacytoma, a Cocker Spaniel was also affected [6].

Interestingly, at the time of the first ophthalmic examination of the conjunctival mass, 2/3 Cocker Spaniels included in the current study were also diagnosed with keratoconjunctivitis sicca (KCS). Lympho-plasma cellular infiltration of the conjunctiva in dogs with KCS is suggestive of an immune-mediated disease [19], and mucocutaneous EMP have been reported to arise at or near areas of chronic immune stimulation [20]. A role of KCS in triggering B-cell proliferation and consequently causing development of EMP cannot be excluded, similarly to what is observed in Sjögren syndrome in humans [21,22,23]. Chronic inflammation has also been related to plasmacytomas in genetically susceptible mice [24]. Specifically, deregulation of IL-6 expression, a cytokine produced by macrophages and responsible for plasma cell differentiation, may be involved in the pathogenesis of plasma cell neoplasia. Moreover, a previously published case report of an intraocular EMP in a cat suggested a possible relationship between the development of the intraocular plasmacytoma and a previous ocular trauma with secondary proptosis and chronic uveitis [8].

KCS in Cocker Spaniels is considered a condition of possible hereditary origin [25]. Therefore, it needs to be determined whether the apparently higher frequency of EMP in this breed is consistent with a genetic/familiar predisposition or results from a higher frequency of dry eye syndrome and secondary chronic inflammation.

The vast majority of primary conjunctival EMPs included in our study had similar clinical appearances: most cases involved the third eyelid conjunctiva, and the tumors presented as smooth, well-defined pink nodules, 2 to 15 mm in diameter. The gross features of the neoplastic lesions were consistent with previous reports of EMPs in humans, dogs, and cats, but they cannot be considered pathognomonic. In fact, at clinical/ophthalmological examination, several differential diagnoses can be made: hemangioma/sarcoma, granuloma, squamous cell carcinoma, mast cell tumor, lymphoma, nodular granulomatous episcleritis, xanthogranuloma and, if the tumor affects the third eyelid, adenoma/carcinoma of the lacrimal gland. In contrast, the histopathological features of plasmacytomas are quite distinctive, and with the exception of markedly anaplastic tumors, diagnosis is usually not difficult and can be made at optical microscope examination. Immunohistochemistry is recommended mainly for more undifferentiated tumors, while, particularly in cats, it is generally considered advisable to exclude systemic involvement, i.e., MM.

According to current literature, the presence of at least two of the following four features is needed for a diagnosis of MM: monoclonal gammopathy, bone marrow invasion with greater than 5% infiltration of plasma cells in sheets or clusters, Bence-Jones proteinuria, or radiographic evidence of osteolytic bone lesions [5]. CBC, serum chemistry profile, serum protein electrophoresis, urinalysis, skeletal radiographs, abdominal ultrasound, and bone marrow aspiration are indicated in cases where MM is suspected.

However, while it is useful in humans and dogs, this protocol could give inconsistent results in cats. Most feline patients with MM, in fact, do not have bone lesions, and monoclonal gammopathy and Bence-Jones proteinuria have also been reported in feline lymphoma. Hypercalcemia, possibly secondary to tumor-induced osteolysis, is frequently reported in humans (25%) and dogs (16%) with MM, but has only been reported in two cats with MM [26].

Lymph-node assessment, bone marrow biopsy, survey radiographs of the spine and proximal long bones, and abdominal ultrasound should ideally be performed in cats to exclude bone marrow plasmacytosis higher than 5%. In the cat included in the present study, the bone-marrow cytological examination excluded MM.

In the present study, a complete diagnostic workup, excluding the presence of MM, was carried out in only 3/7 cases. In the remaining cases, the owners declined further examinations, mostly due to financial constraints.

For a number of reasons, however, it is unlikely that MM was present in our patients. First, MM typically is aggressive and has a rapid course, and its survival rate is low. Second, the results of the clinical examinations and blood assays did not suggest MM. Third, none of its most common clinical signs were reported during the follow up period (i.e., lethargy, anorexia, weight loss, and lameness). Anemia, bone pain, pathologic fractures, renal insufficiency, and recurrent infections were not observed. Moreover, the ophthalmic features of MM can also include retinal hemorrhage and detachment due to hyperviscosity and hyperproteinemia, corneal crystals, ciliary body cysts, and papilledema [27], which were not observed during the course of study. Finally, the previously reported median survival time for cats (170 days) [5] affected by MM is shorter than the follow-up time for the cat being the subject of the study, and although the median survival time in dogs affected by MM treated with chemotherapy (540 days) [28] is longer than the follow-up time, it should be noted that no dogs included in our research underwent chemotherapy.

Canine and feline cutaneous EMP are generally considered benign tumors, with a low recurrence rate after surgical removal [5,17]. In human medicine the recommended therapy for small (<15 mm) or well-circumscribed solitary EMP is typically excisional biopsy [29] in some cases with supplemental cryotherapy [30]. For larger lesions, surgical excision has been combined with chemotherapy or radiotherapy [31].

All the tumors in the present study were surgically removed, and no additional therapy was prescribed after surgery. There was no recurrence in any cases during the follow up time. Thus, the present findings suggest that primary conjunctival EMP is characterized by low local aggression and systemic involvement capacity, and that local surgical excision either without adjunctive treatment or combined with cryotherapy should be curative.

The present study has a few limitations that should be considered: its retrospective design and the relatively small number of patients due to the comparative rarity of EMP led to some variation in diagnostic imaging and follow-up time. Nevertheless, this study contributes valuable information to the little that is known about primary conjunctival EMP in dogs and cats.

## 5. Conclusions

Although it is rarely reported, primary conjunctival EMP should be included in the differential diagnoses of canine and feline conjunctival masses.

Staging recommendations for dogs and cats with conjunctival plasmacytoma should include multiple myeloma diagnostic evaluation. A more standardized approach may allow future studies to evaluate patients for additional prognostic factors.

## Data Availability

The row data supporting the conclusions of this article will be made available by the corresponding author upon reasonable request.

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
