# Peer review of "Conjunctival Extramedullary Plasmacytoma in Dogs and a Cat: Clinical Characteristics and Histopathological Findings"

_animals, 2022, doi:10.3390/ani12213015_

Round 1

Reviewer 1 Report

This is a nicely written manuscript, which will require relatively minor corrections. My suggestions are the following:

Figure 2. Arrow is pointing to the mitotic figure, so this needs to be stated in the figure legend

Page 7. IHC results could be organized in the table. Please use vocabulary "CD20+ positive cells" vs"CD20+"

Discussion

Page , 3rd sentence - MM was introduced for the first time without giving the full terminology (multiple myeloma)

Last paragraph, page 7 - correct Sjorgen's spelling

page 8 - statement on hereditary KCS component in Cocker Spaniels should be supported by the reference

Please cite following reference:  PMID 32435472. 

Human literature supports a notion that conjunctival EMPs are not associated with the systemic disease, so you may want to state that in the discussion and reference it.

Reviewer 2 Report

Overall, this is an interesting paper which should be out there, especially given that there is not much published for ocular plasmacytoma. Please see the comments and questions below:

Simple Summary: ' to affect the ocular and periocular....missing word

Based on our findings....delete 'we recommend that' and ' when examining' and add 'in cases of' canine and feline conjunctival masses, followed by investigations to rule out multiple myeloma.

Given that no multiple myeloma was found in any of the cases, why do the authors recommend an evaluation, based on which findings?

'conjunctival' extramedullary plasmacytoma

replace 'regrow' with recur.

consider should be curative.

Abstract: Methods ...submitted for histopathological 'analysis'

Results: ....were diagnosed in a 9-year-old domestic short-haired cat and six dogs, three..., and a Pinscher, respectively. (with third eyelid mass can be deleted here as it's descrived in the next phrase).

upper eyelid can be replaced with upper palpebral conjunctiva

Five tumors occurred in the third eyelid- implies that five masses developed in one third eyelid. Therefore, I would re-phrase it as such:

EMPs affected the third eyelid conjunctiva in five cases (5/7), followed by the superior bulbar (1/7) and palpebral conjunctivae (1/7), respectively.

replace well-limited with well-defined throughout the paper

replace capacity with 'behaviour'

I am not sure about low local aggression- would probably say does not appear to be locally aggressive.

Materials and methods section:

Re-phrase the first paragraph: I suspect that the authors searched the Pathology database for cases of conjunctival EMPs submitted between 2013-2022, followed by searching their medical records retrospectively. Please re-phrase so that it reflects this.

Immediately after surgical excision, 'the mass' was.. was this fixation- formalin and paraffin= done in the clinic, immediately after surgery? or was the mass submitted to the lab? if so, please rephrase.

The presence or absence of recurrence..during 'the' subsequent re-examinations/recheck examinations/follow-ups. delete controls.

Page 3 bottom paragraph: re-phrase in scientific language- The cat was.. The dogs were...

Page 4 missing bracket

No ocular discharge or bleeding was noted 'in any case'.

Figure 2. cytologic 'smears'- would replace with cytological image following fine needle aspiration? of the mass.

When mentioning tests were performed 'for' five cases should be replaced with 'in' five cases- please see that this is the same throughout the manuscript.

Page 6: the neoplastic lesions were removed 'in' all cases not 'from', 'sparing' not saving.

In case 7 not nr.7. in other places it's n.3. Keep it consistent, case 3, rather than case n.3.

Figure 3: hematoxylin and eosin 'stain': 'the' neoplastic cells.

In the cat,... infiltrating and replacing the third eyelid glandular tissue.

Were the masses removed with clear margins?

Paragraph with CT or abdominal ultrasound. Changes suggestive of multiple myeloma 'were not reported in any case'.

Discussion section:

This case series describes (presents a description doesn't sound ok) clinical and histopathological features of primary conjunctival EMPs- already abbreviated previously.

No need to add beings, enough to say In humans,... or in human ophthalmology,...

Delete 'in turn' and use However, most human...

Paragraph with A single case... primary EMP has previously been reported- note the order.

Sjogren not Sjorgen syndrome

Lymph-node assessment... ..should ideally be performed- rather than 'better'.

Paragraph starting  For a number of reasons... Finally the previously reported.... is shorter than the follow-up time for the cat in the present study. avoid 'our'- no dogs underwent chemotherapy.

All the tumors... there was no recurence in 'any' cases...Thus the present findings suggest that...
